# Uncovering intersecting stigmas experienced by people affected by podoconiosis in Nyamasheke district, Rwanda

**Jean Paul Bikorimana**[1,2]*, **Gail Davey**[2,3], **Josephine Mukabera**[1], **Zaman Shahaduz**[2], **Peter J. Mugume**[4,5], **Papreen Nahar**[2]

**1** Centre for Gender Studies, College of Arts and Social Sciences, University of Rwanda, Kigali, Rwanda, **2** Centre for Global Health Research, Global Health and Infection Department, Brighton and Sussex Medical School, University of Sussex, Brighton, United Kingdom, **3** School of Public Health, Addis Ababa University, Addis Ababa, Ethiopia, **4** Centre for conflict Management, College of Arts and Social Sciences, University of Rwanda, Kigali, Rwanda, **5** Center of Excellence in Biodiversity and Natural Resources Management, College of Science and Technology, University of Rwanda, Kigali, Rwanda

* igira170@gmail.com

**Data Availability Statement:** All relevant excerpts of interview transcripts were made openly available

## Abstract

### Background

Podoconiosis is a stigmatising neglected tropical condition, but the nature of podoconiosis stigma has not been fully explored. There is a growing understanding that the experience of stigma is intersectional, yet no research has been done on this matter in relation to podoconiosis. The aim of this paper is to contribute to the understanding of podoconiosis-related stigma by illustrating how multiple individuals' positionalities intersect to shape the experience of stigma due to podoconiosis.

### Methods

We used qualitative research to explore the experience of stigma among people affected by podoconiosis. Data were gathered using participant observation, interviews, focus group discussions and participatory methods. A total of 59 participants, including patients, local leaders, community health workers, and family members, were purposively selected to participate in this study.

### Findings

We identified three features in which stigma related to podoconiosis was grounded. These were bodily deformation, illness contamination and bodily weakness. The experience of stigma was shaped by the convergence of multiple individual positionalities and identities. Social positionalities and identities based on socio-economic, gender, age and illness status intersected to shape podoconiosis stigma.

### Conclusion

Our study demonstrates that the experience of stigma due to podoconiosis is intersectional, going beyond biological features of podoconiosis. The stigma experienced by affected

in a public repository with DOI: https://zenodo.org/records/13842797.

**Funding:** Funding: JPB, PN and PJM were supported through NIHR contract NIHR200140. The funder had no role in study design, data collection and analysis, decision to publish or preparation of the manuscript. The views expressed in this publication are those of the authors.

**Competing interests:** The authors have declared that no competing interests exist.

individuals is complex, and shaped by the convergence of social structures and many individual identities such as socio-economic status, gender, disability and age. This understanding is crucial to analysing stigma related to podoconiosis, or other NTDs, and for designing interventions that address stigma that arise from biological aspects of disease and social factors simultaneously. Such holistic interventions can significantly improve the well-being of those affected by podoconiosis.

## Author summary

Podoconiosis is a neglected tropical disease that causes swelling of the feet and legs leading to bodily disfigurement. An estimated 4 million people are affected worldwide, while in Rwanda, more than six thousand are affected with the disease. Those affected experience social exclusion and discrimination, which cause profound socio-economic and psychological impacts. The nature of this experience has not yet been fully explored. The aim of this paper is to explore the social nuances of podoconiosis-related stigma to contribute to better understanding of the experience of podoconiosis.

We explored the experience of stigma of people affected by podoconiosis. We identified various features of podoconiosis-related stigma, such as bodily deformation, illness contamination and bodily weakness. We also noted that socio-economic status, gender, age and severity of podoconiosis fuelled the experience of stigma. Participants explained how affected people experienced stigma that was influenced by many factors: their living conditions, age, gender and illness status, on which their social acceptance or rejection was judged.

The experience of affected people is multi-layered, and varies from individual to individual. Therefore, our understanding and actions should rest on an approach that enables us to analyse and identify intersected social features to reduce podoconiosis-related stigma holistically.

## Introduction

Podoconiosis is a neglected tropical disease (NTD) arising from the inflammation and gradual occlusion of the lymphatic vessels, leading to swelling of the feet and legs. It affects genetically susceptible individuals living in impoverished and disenfranchised communities in the tropics and subtropics who spend prolonged periods working and walking barefoot [1]. A recent study suggests the presence of podoconiosis in 17 countries globally [2]. Of these, 12 are African countries. In Rwanda, more than six thousand people have podoconiosis [3].

Those affected experience interlocking health, socio-economic and psychological problems and stigma. While several studies on podoconiosis have focused on the stigma that arises from features of podoconiosis, there is a growing understanding that stigma can arise from multiple sources, more particularly in intersected ways. The central theme in this paper revolves around how social identities or positionality [an individual's location in relation to one's values and various social identities, including class, gender, age, and geographical location.] intersect to shape the experience of stigma associated with podoconiosis.

Studies have shown that podoconiosis is a stigmatising condition. Those stigmatized experience social oppression and disadvantages of various kinds including discrimination, humiliation, denial of employment and social exclusion and rejection [4]. Studies have linked

podoconiosis-related stigma with hereditary and causal beliefs [5–7]. Affected people are excluded and avoided based on the fear of transmitting the disease. Gender has also been shown to influence the experience of stigma. Affected women experience abuse based on failing to fulfil their duties, including lack of productivity [8].

However, despite existing evidence that health-related stigma is intersectional, there is no research has been conducted on podoconiosis and intersectionality [a feminist conceptual framework for understanding how various social identities such as age, class, gender, religion etc intersect to create complex systems of oppression and disadvantages] [9]. Previous studies on podoconiosis explored the various manifestations of stigma from a social-psychological perspective. However, these previous studies have not investigated the intersectionality of stigma with social identities [6,7,10,11]. Lack of research of this nature may result in failing to uncover critical aspects of stigma, hampering the development of effective interventions to combat podoconiosis stigma [12]. Scholars have suggested that stigma can arise from social identities, more particularly from the interconnection of those identities. Certain health problems, particularly NTDs, are associated with other stigmatising features, such as gender, poverty, and inadequate hygiene [13], which are rarely dealt with while managing illness in the clinical setting. Some NTDs like podoconiosis are linked to other stigmatizing features such as scratching, disability, disfigurement, poverty, social status, age, and gender [13,14]. These features form multiple social positionalities and identities that are demeaning and discrediting to those affected.

A number of studies have shown that these social identities intersect with illness or other health features to shape the stigma and oppression those affected experience. Studies on other conditions like HIV/AIDS, schizophrenia, and leprosy have documented the experience of stigma arising from multiple sources. A study in Uganda documented how masculine traits intersect with HIV/AIDs to intensify the experience of stigma. A similar study in Swaziland [Eswatini] reported that men who have sex with other men experience dual stigma which arises from both sexual identity and HIV/AIDs [15]. Another study in Indonesia demonstrated that multiple identities–age, gender, class, religion and disability–intersect with illness to amplify the experience of stigma [16]. The term multiple stigma has been used in these studies to describe the stigma, which arises from many sources.

Social features, including disability, poverty, and gender are are of a great concern among people with podoconiosis [17–19]. This suggests the importance of exploring the nature of the stigma experienced by people affected by podoconiosis.

Theoretical Framework: We used the framework of intersectionality to explore the experience of podoconiosis-related stigma. Intersectionality contends that identities related to socio-economic class, gender, geographic location, and illness status intersect to shape their experiences [9]. The philosophical direction for this study is founded in the belief that human experience is a complex phenomenon, which cannot be explained by a single axis, rather by a plurality of positionalities and identities [9].

Multiple positionalities and identities intersect to shape the everyday experiences of people in a particular situation and context. Those identities are established on societal norms, cultural beliefs and power dynamics [20]. The social identities of people play a role in shaping our ways of living and experiences to our advantage but also our disadvantage [21]. So, for example, social identities such as gender, age, religion and economic class intersect to shape the experiences of people affected by NTDs like podoconiosis. This means that affected people from different social categories may experience podoconiosis stigma differently.

Although studies on podoconiosis and intersectionality are limited, if not absent, the framework has been applied to other health-related conditions [15,16,22,23]. A study in Indonesia

applied intersectionality to HIV/AIDS and other stigmatizing illnesses including leprosy and diabetes [16].

In Uganda, Mburu applied the framework to the experience of people with HIV [22]. The findings of those studies demonstrated how socio-economic positions altered the experience of the stigma the participants faced in their communities.

While some scholars criticize intersectionality for lacking precision in its application, Collins and Bilge argue that intersectionality should be applied as either praxis or an inquiry method [24]. In this study, we used intersectionality as an analytical framework. This allowed us to understand and unfold the complexity of the patients' experiences, not only between social categories but also within social categories [24]. By using intersectionality, our findings shed light on ways in which multiple aspects such as biological features of podoconiosis, social identities, and structures intersect to shape the experience of podoconiosis stigma. Scholars have demonstrated that NTDs are associated with factors beyond illness that are stigmatising [14]. This emphasises the importance of understanding the nature of podoconiosis stigma, and other NTDs by investigating how the intersection of multiple social identities, systems and the aspects of illness contribute to the stigma. Therefore, our study can inform policies and the development of inclusive interventions aimed at reducing podoconiosis stigma that target not only podoconiosis features, but also pay attention to individual identities and social systems that contribute to the experience of stigma.

## Methods

### Ethics statement

This study was approved by the Research Governance and Ethics Committee at Brighton and Sussex Medical School and the University of Rwanda Research Ethics Committee. A permission letter from the district office was taken to the local authority, and access to the community was granted. Participant information sheets were given to the study participants beforehand. A consent form was given to each participant who voluntarily agreed to participate for signature. JPB kept one copy of the signed consent form and the participants kept another copy. Issues related to power dynamics between research and participants, insider and outsider identities, and emotional trauma from hearing painful stories of patients were experienced and dealt appropriately.

### Study design

We employed a qualitative research design to explore features of podoconiosis that attract stigma, and how multiple positionalities shape the experience of stigma among people with podoconiosis.

In-depth understanding of a phenomenon under study is crucial in qualitative research [25]. Qualitative design helps to understand how people make meaning of their experience, and make sense of their social context. It allows researchers to interpret people's meanings while remaining grounded in their perspectives [25].

### Setting and study participants

This research was conducted in Karengera sector in Nyamasheke district in Western Rwanda. The district holds the highest number of podoconiosis cases in the country. [3] The district has a total population of around 381,800, who are primarily subsistence farmers [26]. It is a mountainous rural area, with poor roads and street networks. The population of Nyamasheke is predominantly female (53.3%) and power relations between women and men are based on

patriarchal principles. More than 62% of the population are aged less than 25 years old, while the percentage of the population aged greater than 60 years is relatively low (5.6%). There is one health centre and two health posts in Karengera, but people seek healthcare from traditional healers as well. There is a wide range of denominations, including Catholic, Protestant, Seventh Day Adventist, Pentecost, and Jehovah's Witness, which are followed by 98% of the population. The census also reported 33 people affiliated with traditional religion in Nyamasheke district [26]

Participants for this study were all adults (18 years old +) and included affected people, community representatives, and community health workers. CHWs are community members who volunteer to provide basic health care services for health problems and health education. Non-affected community members were recruited to gather rich data regarding social context and cultural understandings in relation to podoconiosis. Participants were purposively selected and maximum variation sampling principles guided recruitment so that diverse characteristics such as gender, age, duration of illness for patients, education status, religion, and wealth categories (*Ubudehe* in Kinyarwanda) were represented. The principle of data saturation was used to determine the number of participants [27].

During the first phase, we used a list of podoconiosis patients provided by Rwanda Biomedical Centre, and ten participants were recruited. The second phase concerned "perceived" patients [this refers to individuals with podoconiosis selected via other patients, who were asked whether they knew a person with the same disease], and six patients were recruited. This helped us to learn more about how individuals affected by podoconiosis are perceived in the community. Participants were selected one at a time, allowing time to adjust the interview guide to get more understanding as needed. Those affected were visited in their homes by JPB and his field guide.

Informants and family members of patients were selected based on the stories of participants with podoconiosis. Narratives contained stories about the involvement of members of the community, including leaders and service providers, who played a role in the lives of those with podoconiosis. These stories were useful in selecting the right family members and KIs.

CHWs were recruited with the help of the person in-charge at Mwezi health centre [28]. The person in-charge of CHWs distributed the information sheets to thirty-two CHWs and arranged appointments with those agreeing to participate. Before the discussion started, the aim of the study was reiterated, and each participant gave written consent.

## Data collection and tools

The fieldwork ran from March to December 2022. Data were collected using mixed qualitative methods, including participant observation, interviews, and focus group discussions (FGDs). An observation checklist was used, however observation was made beyond the checklist as various scholars suggest researchers focus, and simultaneously provide details about what is being observed [29]. Checklist items included prominent activities, division of labour between men and women, behaviours and practices, home environment, social interactions, people and object interactions, and power dynamics. During the visit, questions were asked in the form of natural conversation, and notes were taken. Field notes were also taken [30].

An interview guide was developed for in-depth interviews with patients and their family members. Flexibility around the guide helped to explore participants' perspectives more deeply. For key informants and the FGD with CHWs, interview guides were used and covered questions related to general understanding of social structures, cultural beliefs, the health care system, stigma and discrimination, and podoconiosis. Interviews and FGDs were conducted in a local language, Kinyarwanda, and audiotaped using an audio recorder device. Techniques,

for example, probes and prompts that encouraged study participants to generate in-depth data were employed [31,32]. Recordings were played several times and important questions written down to be explored in subsequent interviews.

## Data analysis and trustworthiness

Audio data were transcribed into text-based data, and were analysed thematically [33]. Transcripts were imported into NVivo version 20 to organise and code. The transcripts were kept in the language of the interviews, Kinyarwanda, but the coding process was done in English, and quotes were translated into English.

Data were read line by line several times, and memos were written to allow comparison of data from different sources. All data sources–field notes, interviews, focus groups, and transcripts–were triangulated by checking for similarities, differences and contradictions [34,35]. The data were coded both deductively and inductively [36,37]. A code list was generated, and the themes and patterns were developed [33]. Codes were organised and categorised under themes. Then, themes were described and interpreted analysing their significance and meaning in relation to the experience of podoconiosis. Interpretations of the key themes were crossed-checked among participants and revisions made accordingly [35].

## Results

This section contains the findings of this study, and has two sub-sections. 1) Demographic characteristics to give an overview of the study participants. 2) Three main sources of stigma; bodily deformation, fear of contamination and bodily weakness. Each theme will contain ways individual identities intersect to shape the experience of podoconiosis stigma.

### Demographic characteristics

A total of 59 participants took part and included sixteen women and men with podoconiosis, five of their family members, three local leaders, one community church leader, one healthcare professional, Community Health Workers (CHWs) (32) and one traditional healer. Patients' ages ranged from 25 to 90 years old, and a large number (13) had podoconiosis for more than ten years. The majority of patients (11) were female, of whom four were married, three were separated from their husbands, two were widows, and two were unmarried. Most men were married. Half of the participants with podoconiosis had no formal education. Majority of the participants were farmers, and more than half were categorised in the first (poorest) wealth category.

Most of the key informants were men, and aged between 35 and 63 years old, and the 16 female and 16 male CHWs were aged between 35 and 62 years old.

### Bodily deformation

Disfigurement of the body due to podoconiosis compounded with other features causes affected people to be stigmatised. Bodily deformation-related stigma is grounded in physical appearance. Most participants confirmed they had experienced stigma because of their enlarged feet and legs. They linked social prejudice, discrimination and exclusion to their disfigured and swollen legs. They said that because of the large size of their feet and legs, their walking gait was impaired and their physical appearance distorted. Gender, duration of illness, age, and economic status influenced the experience of stigma due to bodily deformation. Gender and duration of podoconiosis were the most prominent identities shaping stigmatisation,

and they worked together to fuel stigma for middle-aged poor women. Women were more concerned about the deformity of their feet and legs than men were.

Based on participant observation, the shape of women's bodies was considered an important trait governing their social acceptance. KIs also confirmed that the beauty of women was valuable in the area, and it was one of the determinants of acceptance and value for women. Most participants reported that women with podoconiosis experienced social disapproval because of their distorted beauty. Only one man reported being concerned about his physical appearance and that it was difficult for him to find trousers for his swollen legs. Participants said that the deformities of their feet and legs were linked to the stigmatising names they were called, including *Nyirakiguru* [refers to a woman with swollen legs], and *Birenge* [the one with enlarged feet]. These terms were mainly attributed to women with podoconiosis.

*"The reason they (people) are scared about them is because of the appearance of their skin, people with that illness have frightening skin, and they isolate themselves". (Key informant, M, 53 years old)*

*"It is not a big problem for men, [but] can you imagine seeing a woman with unequal sized legs, it is shameful, and people stigmatised you just because you have swollen legs". (Patient, F, 60 years old)*

Duration of illness fuelled the experience of stigma. Participants said that those with severe podoconiosis had a frightening appearance, and that this fuelled social disapproval for those with severe illness. This source of social disapproval frightening was a concern for working-aged women. Based on participant observation, most women who had had podoconiosis for a lengthy period had large, dirty feet and legs, and rough skin. Most of them were living in poor conditions, such as old houses, had difficulty in getting water, and experienced shortage of food. Participants linked these conditions with inadequate hygiene, scratching, smelling, and exacerbation of their condition.

Participants reported that scratching, unpleasant odour, and inadequate hygiene intensified the stigma for people with podoconiosis, and were associated with poverty. They linked scratching with inadequate hygiene due to the skin changes that often occur after scratching. Participants mentioned that poverty was not stigmatised *per se* but was linked with negative attitudes that attracted social prejudice. They said that poverty fuelled stigma due to the disfigurement of the body by exacerbation of bodily deterioration, which participants attributed to a lack of proper hygiene, and inadequate resources to seek help or afford sanitary materials. Participants talked about how poverty and the features of podoconiosis were reasons for stigmatisation of those affected.

*"The reasons are, one let us say that you are a rich person…many people with ibidido [greatly swollen feet and legs], they have problems of poor hygiene, for rich persons, their houses are clean, and they look clean too, and they take care of themselves…trying various "arrangements" for treatment". (CHW, M, 45 years old)*

*"When it (the foot) leaks, it produces bad odour, and nobody would come close to you because of the odour…they avoid you…why because I used to work and earn some income to buy soap, lotion etc. but I cannot do it anymore…how should I get soap"? (Patient, F 45 years old)*

*"Poor people are not stigmatised in this area, but when this poverty became worse, people ignore and avoid them particularly for those who do not care for themselves". (Key informant, M, 42 years old)*

However, one woman with severe podoconiosis reported that she was no longer being stigmatised because her neighbours had become familiar with her, and the members of the community knew her condition.

*"People in this village know my condition, and we live peacefully they do not discriminate me because of my legs". (Patient, F, 61 years old)*

These findings indicate how the established podoconiosis stigma transcends having disfigured feet and legs, rather it intersects with social position and multiple individuals identities such as gender, socio-economic status, severity of disease. Cultural standards, expectations and norms attached to the groups of individuals according to one's social membership contribute to the stigma they experience. These findings are crucial in informing the analysis of the nature of stigma related to bodily deformation, and development of combined interventions that address physical deformation while taking into account gender differences, socioeconomic status, age groups and illness stages leading to a holistic approach to alleviate stigma.

## Fear of contamination

Explanatory beliefs held around podoconiosis contributed to the stigmatisation towards affected people, and class, gender, severity and duration of podoconiosis shape the experience of stigma. Terms used to describe podoconiosis in the area and causal beliefs, particularly hereditary beliefs were linked with social prejudice and exclusion arising from the fear of contracting podoconiosis. Terms were constructed based on the visibility of symptoms, and the amount of swelling shaped the stigma affected individuals experienced. Participants talked about stigmatisation that arose from fear of contracting the illness because of physical appearance.

Most participants linked the stigmatisation of those affected to the belief that the illness could be transmitted by physical contact. Duration of illness and poverty fuelled the experience of stigma by the visibility of symptoms. Poor affected people with severe symptoms were avoided for fear that the illness could be contagious because of symptoms, which were thought to be worsened by inadequate hygiene due to lack of the capacity to afford buying sanitary materials.

*"They (people) fear to contract it; they think that they can get it when they physically contact us, because my legs looks frightening, with many cracks and wet open wounds". (Patient, F, 60 years old)*

*"Even adult people are scared when they see the legs of those people because they do not have knowledge about the illness. Sometimes it is confused with leprosy which is contagious". (Key informant, M, 35 years old)*

*"I had a friend. . . I shared with her how I wash my mother's wounds, and next day she refused to share meals with when I invited her as we used to share meals before I told her my story". (Family member, F, 28 years old)*

Hereditary beliefs was commonly considered a source of stigmatisation based on the belief that the disease runs in families. The belief that podoconiosis could be inherited intersected with other features as a source of stigma, and women bore a huge amount of stigma around marriage. Participants talked about how men did not marry affected females or those from podoconiosis families to avoid having children with the same illness. Transmitting podoconiosis was not only a source of stigma perpetrated towards those affected but also a source of self-

stigmatisation affecting both men and women. Some participants reported that they felt ashamed of being a source of problems in their families, based on the fear of transmitting both illness and stigma to their children and grandchildren. An old woman described how she had been the cause of the breakup between her daughter and her fiancé. She talked about how her daughter had brought her fiancé to meet her mother, and the man left the girl because the illness runs in the family, and they feared that one of their descendants would have the same illness in the future.

*"I remember when my daughter brought her fiancé to visit us, but they broke up afterwards and I thought I was the cause". (Patient, F, 60 years old)*

*"My daughter gave a birth to a child with congenital deformities, and I did not sleep on that day because my husband became angry at me, accusing me of being the cause". (Patient, F, 48 years old)*

*"Some men may fear marrying a woman with that illness because they thought the illness runs in the family and fear to have children with that illness". (Key informant, M, 48 years old)*

These findings provide valuable perspectives around how explanatory model-related stigma intersects with social prejudices arising from individual identities to shaped podoconiosis stigma. The experience of stigma that arise from the fear of contamination goes beyond the perception and beliefs held by community members, but it intersects with individual identities such as gender, marital status and age-related stigma to multiply it because of cultural meanings and values that dictate requirements for group members. The findings imply that measures to combat stigma should transcend awareness-raising alone, but combine interventions to alter understanding of affected individuals about the disease, address gender inequalities, halt the progress of illness and ensure equally-shared resources. This inclusive approach may promote the social integration and well-being of those affected.

## Bodily weakness

Participants associated the bodily weakness caused by podoconiosis with stigmatisation. Based on participant observation, living and working in the area demanded physical strength, and community members were involved in physically demanding tasks on a daily basis. KIs also confirmed that living in the area required bodily strength for people to work hard to earn a living. Activities undertaken in the area included working in fields, carrying manure, making, and transporting bricks, and fetching and carrying grass for domestic animals.

Gender, employment, education and age intersected to fuel the experience of stigma due to bodily weakness. Both women and men experienced stigma due to bodily weakness, which was shaped by individuals' positionality in the community. Participants talked about how men were expected to be involved in activities that demanded bodily strength and so men were expected to be physically strong.

Based on participant observation, stigma related to bodily weakness among women was not only based on their ability to be involved in activities requiring physical strength. This stigma also arose because women were expected to contribute to the development of the family through income generating activities. Affected women experienced the feeling of being useless to their husbands, and children.

*"He divorced me, he gave a red licence saying go, you disabled woman, because you will not be able to do farm activities, you are unable to do anything". (Patient, F, 60 years old)*

*"I changed places, I live in the other annex that is behind there. . .I do not live together with her (his wife) she always insults me, because she wants me to go to work but it is hard for me to work with my condition. Other people travel far away to look for various jobs, like digging or labour jobs at construction sites, and people want strong workers, and they told me that I do not have strength".* (Patient, M, 68 years old)

*"Men do those heavy tasks, we [women] are involved in business, but we cannot lift heavy loads as men. They [men] carry heavy bunches of banana, it is shameful for a man who cannot do those heavy tasks, and people may laugh at him when carrying, like, one banana".* (CHW, F, 58 years old)

Duration of illness and unemployment fuelled the experience of stigma. Participants linked the duration of the disease with the severity of illness increasing weakness. Those who had podoconiosis for a long time reported being perceived as weak and denied jobs based on the perception of being unable to work. Unemployment was linked in peoples' minds with dependence, begging or theft intensifying stigma for those with podoconiosis. Participants reported that these factors were highly unwelcome in the community and linked with social prejudices. Observation data showed that types of employment in the area included construction activities and farm work, and these were mainly reserved for those who were not employed in the formal sector. Some people were employed in tea gardens, others at small-scale coffee production stations, while a small number of young people were employed in the hospitality sector, including in bars. Many other community members were public servants and worked in education, local administration, or health. People employed in these categories were a mixture of people from the area and other districts, and they were required to have a certain level of education.

*"If you do not have something to do in this area, they will tell that you want to be a beggar or a thief".* (Patient, M, 55 years old)

*"People think that those who do not work will become thieves, because they do not have other means of living".* (Key informant, M, 58 years old)

*"A person who does not work is always a burden to the society. That is how it is in our area. Those who are always at home doing nothing are seen as a problem for their families and even to the society".* (CHW, F, 61 years old)

Employment intersected with education to shape the experience of stigma, so that stigma was intensified for those who were unemployed and uneducated. Most affected participants reported that they had not attended formal education, and education was linked with respect, the ability to speak out in the public and work opportunities. Participants reported that because they had podoconiosis, and no education, they could not get jobs in public services, fuelling the experience of social rejection and disrespect.

*"For example, at the health centre, they may let teachers pass first because they are educated, the one with no education would be treated afterwards. So, there is a difference between those people. In addition, even their thoughts differ because all things emanate from the head. When things in the head have already died, all things are dead. . .. Hum".* (CHW, M, 50 years old)

*"Educated people are respected, for example men pay much money for dowries for educated girls in comparison with uneducated ones".* (Key informant, M, 58 years old)

*"There is huge job opportunity for educated people, and if I had had education, I would not have been like this and even my husband would have chased me because I would have been doing something to generate money for our family". (Patient, F, 60 years old)*

Education appeared to be valuable for employed persons. Some participants talked about their understanding around education, that education was valued based on the outcome, which was employment. They reported that education alone was not valuable except for those who became employed, who were valued and respected.

*"Educated people who have jobs are more valued and respected than those with no employment. They (working educated people) are considered essential to their families as well". (Key informant, F, 49 years old)*

*"Educated people, particularly when they got jobs, are respected and are considered rich because they have a monthly salary". (Key informant, M, 35 years old)*

*"Education level alone does not mean anything, you need also to have an employment; otherwise, you are like other people who have not gone to school". (Key informant, M, 58 years old)*

Age intersected with other factors, including duration of illness and employment, to shape the experience of stigma. Based on participant observation, people of working age were expected to work hard, while this was not true for the elderly, who were less concerned with respect to employment.

Unemployment for working-aged people was associated with stigmatising attitudes. Age intersected with duration of illness, gender, and employment to shape the experience of stigma for people with podoconiosis. Participants said that those affected were perceived to be beggars or lazy people because they did not work, fuelling stigma for unemployed middle-aged participants. Young girls were concerned about marriage based on the fear that their condition would worsen, affecting their working ability, which was considered for marriage.

*"Younger individuals are stigmatised, but me, you see my body is full of wrinkles and nobody is any longer interested in me". (Patient, F, 61 years old)*

*"Young people experience many problems, people pity them saying that they will not be able to work for their future but for elderly people sometimes people relate it with aging illnesses". (CHW, M, 65 years old)*

*"Young girls are more affected, they face more problems compared to young men, and a man does not marry a girl with podoconiosis. But for us old people, we are no longer concerned about given that we are old". (Patient, F, 60 years old)*

Adults admitted that affected young girls experience more stigma than elderly and their counterpart boys with the same condition do. More attention was paid towards girls, and they were pressurised to get married to preserve the dignity of the family.

These findings indicate how social identities and systems such as education, employment, gender and age contribute to the podoconiosis stigma beyond the stigma related to bodily weakness. Contextual factors such as rurality, patriarchy system, mountainous landscape, and agricultural dependence contribute to the perception and expectation of group members, thus multiplying stigma due to diminished working and walking ability. Understanding of the stigma related to bodily should not be separated from the social prejudice and exclusion caused by individual identities and social positionalities. This understanding is of importance in

informing planning and development of interventions to combat stigma. It suggest the combined interventions that promote positive bodily appearing and strength, job opportunities and gender equality simultaneously.

## Discussion

The nature of podoconiosis-related stigma remains under-researched in Rwanda. To our knowledge, this is the first study to explore podoconiosis-related stigma in the country, more particularly through the lens of intersectionality. Our findings reveal that the stigma experienced by individuals affected by podoconiosis is not solely due to the condition itself. Rather, the sources of stigma are multifaceted, rooted in various positionalities and features of the illness, such bodily deformation, weakness, and beliefs around the nature of contamination. This experience of stigma is ultimately shaped by the individual's positionality.

We identified various features–bodily deformation, illness contamination, and bodily weakness—on which stigmatisation towards podoconiosis was grounded. Each intersected with multiple positionalities to shape the stigma experienced by affected people. Affected people experience stigmatisation from a combination of people's social positionality, compounded by features of podoconiosis.

These findings are consistent with those of studies on other stigmatising conditions that have documented the intersectional experience of stigma by affected people [16]. Similar to previous studies, our findings challenge the understanding of podoconiosis-related stigma as limited to features of the disease such as heredity, causal beliefs, or contagiousness [5,38]. This paper contributes to a holistic understanding of stigma due to podoconiosis, demonstrating how podoconiosis features intersect with positionality and identities to intensify podoconiosis-related stigma.

The following sections comprise discussions about how each feature intersects with multiple individuals 'positionality to shape stigma.

Podoconiosis stigma due to disfigured feet and legs is intensified by multiple individual positionalities. Stigma due to the deformation of feet and legs intersects with the oppressed positionality people carry to shape the experience of stigma. Gender, socio-economic class and age were found to be crucial in relation to the disfigurement due to podoconiosis. Cultural understandings, norms and societal standards that surround gender play an important role in fuelling podoconiosis stigma. In Rwanda as in different cultures, norms are attached to women and men that guide their ways of behaving and acting [39,40]. Those who cannot meet these norms because of podoconiosis are subjected to social judgement.

Previous studies on stigma have linked disfigurement issues with social acceptability [41], which may be attributed to social standards of acceptance of different categories of people in terms of body image. Those who could not maintain those expected standards due to podoconiosis faced societal prejudices, with profound psychological effects. Previous studies on stigma have documented psychological, economic, and emotional impacts caused by stigma that can result in poor health outcomes for stigmatised people [14,38,42]. A study conducted in Rwanda has documented symptoms of depression among people with podoconiosis [43].

Poverty also fuels the experience of stigma due to podoconiosis in various ways. Firstly, being poor was attached to stigmatising attitudes such as smelling bad (due to poor hygiene), being perceived as a thief or a beggar (based on perceptions of bodily weakness). Secondly, poverty can attract stigma directly, but also it contributes to worsening disease status because of the inability to halt the progression of the disease due to lack of resources for help seeking or self-care. This creates other stigmatising features, including further disfigurement, scratching, and bad smells that attract flies, leading to the experience of multifaceted stigma.

These findings are consistent with previous studies on health-related stigma, which have demonstrated that NTDs are associated with features that may intensify stigmatization [14]. Poverty is a prominent factor that contributes to the miserable lives of people affected by NTDs. Previous studies have linked podoconiosis with poverty, and shown how people can be disrespected and ignored by community members [17]. Scholars have associated being poor with various forms of social exclusion [44]. Poor people are often ignored and disrespected, leading to discrimination and isolation. These findings are consistent with previous studies that have connected poverty and social identities with the adversities people experience and their impacts on health outcomes [13,14]. This underscores how poverty-related identities and illness features intersect to intensify stigma.

Unlike a study in Ethiopia that documented no association between age and the experience of podoconiosis stigma, our findings demonstrate how age shapes the experience of stigma. Age fuels the experience of stigma by interacting with poverty and gender. Disfigurement and disability attract different social prejudices at different ages. While elderly people are not concerned about the deformation of the body, working-aged people experience stigma arising from bodily deformation based on the grounds of prospects and social approval. Elderly people appear to be content with the fact that they have lived their days, and the deformation of their bodies is attributed to the ageing process, which minimises the degree of stigma. These findings contrast with the findings of previous studies that have documented the experience of stigma among elderly people with disability [45,46]. Those findings associated older age with the experience of stigma. Such a difference may due to cultural differences in regions.

In Nyamasheke, like in other parts of Rwanda, elderly people are culturally to be respected and taken care of. The elderly people with podoconiosis do not feel stigmatised. Although this may sound positive, it may have an impact on actions to improve their conditions. A study on coping strategies has demonstrated that affected people actively adopt practices and actions to improve their stigmatising conditions [11].

Consistent with previous study studies [5], stigmatisation based on the fear of contracting podoconiosis has been reported. Our findings contribute to this understanding by adding that stigma is fuelled by the severity of illness and gender. Avoidance perpetuated towards those affected is fuelled by the visibility of symptoms, which is linked to the severity of podoconiosis. The frightening look of symptoms suggests the danger of podoconiosis, thus increasing the fear of contracting the illness, leading to stigmatisation. This form of stigma can be amplified by one's gender, specifically based on marriage acceptability. Similarly, previous studies in Ethiopia have reported difficulties in marriage among affected people [47]. People refuse to marry people with podoconiosis or people from families with podoconiosis, and women are more affected in this regard. This indicates that this form of stigmatisation is based on both fear of contracting podoconiosis and gender.

Bodily weakness is another dimension of stigma related to podoconiosis. As documented in a study in Ethiopia on intimate partner violence [8], affected women experienced various form of abuse, including forced divorce by husbands when their ability to work diminishes indicating gender disparities in the experience of stigmatisation. Our findings show that experience of stigma due to bodily weakness is shaped by the intersection of gender, employment, education and age.

Unemployment and lack of education also attract social prejudices. Similarly, a lack of education and unemployment have been linked with stigmatising attitudes [13]. People in these categories commonly have low social status, unskilled and incapable. Opportunities for manual labour in the area are based on age and physical strength, while skilled jobs are based on educational attainment. Elderly individuals with podoconiosis are not involved employment due to the debilitating effects of podoconiosis and old age. They are denied jobs because of

lack of physical strength and education, leading to a loss of certain social status. This loss of social status puts those affected a lower position; they are regarded as unskilled or incapable, and deprived of jobs, therefore becoming trapped in a cycle of poverty [48].

On the other hand, education was linked with respect based on work opportunities. Education acts as one of the determinants of social position [49]. Podoconiosis affects schooling due to poverty and stigma, contributing to unemployment. These findings are similar to those of a study in Ethiopia that reported discontinuation of schooling because of stigma [38]. Discontinuation of education due to podoconiosis contributes to low social status for those affected, and this can lead to feelings of embarrassment, anxiety, social exclusion and discrimination. Also, those affected linked their unemployment with being perceived as weak, which was linked with stigmatising assumptions such as being lazy, thieves or beggars, again leading to low self-esteem, and social exclusion.

Gender played a significant role in relation to bodily weakness. Women are more often assigned the roles of nurturing and caring that are culturally attached to them, and are expected to help men to contribute to the economic development of the family. The economy in the Nyamasheke is mainly based on agriculture-related activities, which is the main economic activity in the Nyamasheke. People need to have bodily stamina to work in farms. When this expectation was constrained by podoconiosis, the social status of those affected was reduced. This had a profound effect on the well-being of vulnerable people with podoconiosis. This tells us that the experience of stigma due to bodily weakness is intensified by a web of intersected features.

These findings reflect the work of many scholars who have demonstrated how people's identities shape their daily lives. Individual identities and positionality attached to those affected intersected with podoconiosis features. A framework that combines multiple sources can be useful in unpacking nuances of stigma due to podoconiosis (Fig 1). The framework of intersectional experience of podoconiosis in this paper (Fig 1) illustrates how many features of stigma intersected to fuel the experience of podoconiosis related stigma. It contains features of podoconiosis and social identities/positionalities, which all intersect to fuel the experience of stigma.

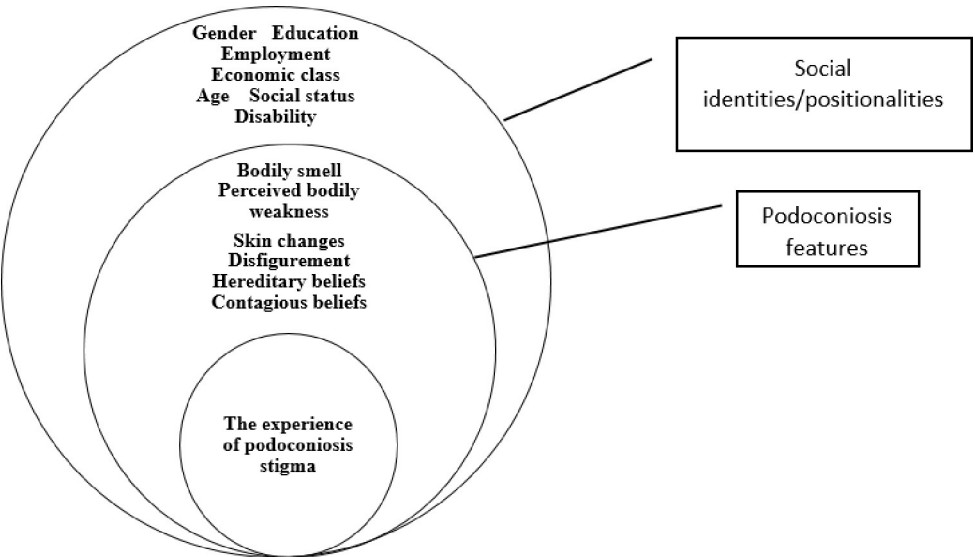

**Fig 1. The framework of the intersectional experience of podoconiosis-related stigma.**

As Crenshaw argues, the oppression people experience is shaped by intersecting axes. This logic is also true in the experience of stigma due to podoconiosis. Affected people face a complex experience stigma that arises from many interlocking sources. They are assigned to diverse identities, including socio-economic status, gender, education level, employment status, and age, and these identities form different positionalities, which intersect with biological changes due to podoconiosis to shape the experience of stigma. This understanding can help in unfolding and understanding the nuances of podoconiosis stigma.

Our study has some limitations. The study's findings are specific to the context in which the research was conducted, which includes particular cultural, socio-economic, and geographical factors. As such, the findings may not be directly transferable to other settings with different social dynamics and health challenges. Also, the observation time was short, and this may have prevented us from observing more realities in relation to the experience of stigma. Future studies may consider conducting similar studies in in different geographic and cultural settings to have a broader understanding of stigma operates across various environments. These studies might consider longer observation times, which would help in unpacking more nuanced realities about the nature and dynamics of podoconiosis related stigma.

## Conclusion

Our study has demonstrated that the experience of stigma due to podoconiosis is intersectional, being shaped by individuals' multiple social positions and identities, which intersect with distinctive features of podoconiosis, thereby amplifying the overall experience of stigma. This suggests that it may be useful to analyse the experience of stigma due to podoconiosis using the framework of intersectionality. This understanding can inform policy makers and help implementers plan for integrated systems and interventions that not only consider podoconiosis as an illness, but also take into consideration identities based on socio-economic status, gender, employment and age assigned to those affected.

## Acknowledgments

We thank all study participants for the time to share with us their experiences. In addition, many thanks go to local authorities and community members for their support and friendship that made this study happen.

## Author Contributions

**Conceptualization:** Jean Paul Bikorimana, Gail Davey, Josephine Mukabera, Zaman Shahaduz, Papreen Nahar.

**Data curation:** Jean Paul Bikorimana.

**Formal analysis:** Jean Paul Bikorimana, Papreen Nahar.

**Funding acquisition:** Gail Davey, Zaman Shahaduz.

**Investigation:** Jean Paul Bikorimana.

**Methodology:** Jean Paul Bikorimana, Gail Davey, Josephine Mukabera, Papreen Nahar.

**Project administration:** Gail Davey, Zaman Shahaduz, Peter J. Mugume, Papreen Nahar.

**Resources:** Jean Paul Bikorimana, Gail Davey, Josephine Mukabera, Papreen Nahar.

**Software:** Jean Paul Bikorimana, Papreen Nahar.

**Supervision:** Gail Davey, Josephine Mukabera, Peter J. Mugume, Papreen Nahar.

**Validation:** Gail Davey, Josephine Mukabera, Zaman Shahaduz, Peter J. Mugume, Papreen Nahar.

**Writing – original draft:** Jean Paul Bikorimana.

**Writing – review & editing:** Gail Davey, Josephine Mukabera, Zaman Shahaduz, Peter J. Mugume, Papreen Nahar.

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
