## [Decision Letter · Decision Letter 0]

1 Sep 2024

Dear Mr Bikorimana,

Thank you very much for submitting your manuscript "“When you have this disease, and you have no money, nobody can respect you”: Uncovering the understanding of podoconiosis-related stigma in Rwanda." for consideration at PLOS Neglected Tropical Diseases. As with all papers reviewed by the journal, your manuscript was reviewed by members of the editorial board and by several independent reviewers. The reviewers appreciated the attention to an important topic. Based on the reviews, we are likely to accept this manuscript for publication, providing that you modify the manuscript according to the review recommendations. 

Please read and carefully consider each of the suggestions from the reviewers, and prepare and submit your revised manuscript within 30 days. If you anticipate any delay, please let us know the expected resubmission date by replying to this email. 

Sincerely,

Bruce A. Rosa

Academic Editor

Victoria Brookes

Section Editor

Reviewer's Responses to Questions

**Key Review Criteria Required for Acceptance?**

**Methods**

-Are the objectives of the study clearly articulated with a clear testable hypothesis stated?

-Is the study design appropriate to address the stated objectives?

-Is the population clearly described and appropriate for the hypothesis being tested?

-Is the sample size sufficient to ensure adequate power to address the hypothesis being tested?

-Were correct statistical analysis used to support conclusions?

-Are there concerns about ethical or regulatory requirements being met?

Reviewer #1: Yes

Reviewer #2: The introduction gives a good account of the issues underlying stigma, and in particular justifies the need for the research to be undertaken, and the intersectional approach used. The importance of positionality is well described, including a well referenced summary of the key findings in literature to date. This level of nuance in thinking around stigma and NTDs is welcome, and makes a good case for the research methods used and the theoretical framework applied. 

While the list of factors with some evidence of relevance to stigma is comprehensive, I think explanatory models (around causality) are particularly key and could be expanded upon a little. These of course intersect with different characteristics of individuals and groups, reinforcing stereotypes mentioned. Religion and different approaches to health care (traditional vs modern/western) are mentioned, and this would be of key interest, not only because of different ideas of causality that populations might have in relation to the impact on stigma, but also because it would affect design of both stigma interventions, and help-seeking behavior in general.

The methods, sampling etc are very appropriate for the study, and it makes sense for the study to be carried out in an area with high podoconiosis prevalence. Sampling is well described (though some more detail around what constituted a 'perceived' patient would be useful).

Ethical approvals are well described and overall in the paper there is a welcome sense of how the global cooperation worked. It is useful to go beyond the description of ethical approvals/consent process, to also explore what some of the ethical issues that might be expected would be, and describe how these would be addressed in the methods and engagement with participants.

While the Author's Summary gives a good lay description of the study, I think there would still be value in explaining the meaning of words like intersectionality and positionality when first used. This very brief primer would help some readers have a better insight into why these are important. While recognising that the research referenced was probably carried out prior to the name change, I suggest you add Eswatini/eSwatini in parentheses after the mention of Swaziland.

Reviewer #3: The objectives of the study clearly articulated 

The study design is appropriate to address the stated objectives

The study population is clearly described 

Were correct statistical analysis used to support conclusions

Ethical issues are well addressed

**Results**

-Does the analysis presented match the analysis plan?

-Are the results clearly and completely presented?

-Are the figures (Tables, Images) of sufficient quality for clarity?

Reviewer #1: Yes

Reviewer #2: The results are comprehensive and rich, and it is good to see the reference to where intersections occur, which is laid out in a good level of detail. 

In general they are in keeping with established findings, but add some valuable detail, that would be of value to implementers in this area. 

The use of quotes is effective and adds colour to the results. Please translate 'ibidido' (line 305, page 13), which I assume means 'podoconiosis' but may contain more subtle meaning that would be useful for a non-Kinyarwanda speaker to understand.

Reviewer #3: The analysis presented match the analysis plan

The results are clearly and completely presented

The figures are of sufficient quality for clarity

**Conclusions**

-Are the conclusions supported by the data presented?

-Are the limitations of analysis clearly described?

-Do the authors discuss how these data can be helpful to advance our understanding of the topic under study?

-Is public health relevance addressed?

Reviewer #1: Yes

Reviewer #2: The results described are followed through well in the discussion, but I wonder whether a table or graphic of some sort might make the learning more accessible and systematised. The figure on page 26 is useful, but quite high level, and some more details around how the identities and stigmatising factors intersect might be possible to represent graphically. This is an idea, and may or may not be considered appropriate or useful by the authors.

Under 'limitations', the authors correctly refer to the results being contextual. It would be helpful to hear their thoughts on either how the actual learning from this area might general elsewhere, or how the methods and approach might be applied elsewhere so that this research and methodology can inform work in other locations.

Reviewer #3: The conclusions is supported by the data presented

The limitations of analysis are clearly described

The authors discuss how these data can be helpful to advance our understanding of the topic under study.

The public health relevance are addressed

**Editorial and Data Presentation Modifications?**

Reviewer #1: Minor comments

1. Either the short title or the full title needs revision. The two titles seem to convey different meaning. The core theme of the study is "intersectionality" of podoconiosis stigma. I would suggest the title to be revised as "...: Uncovering the intersectionality of podoniosis-related stigma in Rwanda and the short title to read as "Intersectionality of podoconiosis-related stigma..."

2. In the abstract, the last statement in the conclusion section is incomplete. The reader may ask, "To achieve what? What are the implications of this conclusion to the broader scientific audience, policy makers and implementers?"

3. The statement on line 99 is not true. Studies conducted by Tora et al, 2011, 2012 and 2014 and Ayode et al. 2013 and 2016 explored the various manifestations of stigma from social psychological perspective. The authors can articulate their points that "previous studies have not investigated the intersectionality of stigma with social identities". 

4. On line 126 and 127, the phrase "identities based sociodemographic ..." is confusing. Revision is needed here. 

5. On line 138, the word "illnesses" should be replaced by "stigma". 

6. Lines 146-150, needs revision including the relevance of this study to NTD researchers, policy makers and implementers. 

7. On line 207, "attitude" is included in the observation checklist. How is it possible to observe attitude? 

8. The findings presented demographic characteristics beginning from line 258 should be categorized under an overarching theme. I suggest general themes like 'causes of stigma' or 'sources of stigma'. 

9. Why did the authors didn't address the 'intersectionality' of the sources of stigma with social identities such as age, gender, economic status in separate sections?

Reviewer #2: I have mentioned a possible representation idea in 'Conclusions' above.

Reviewer #3: "Minor revisions"

Page 4, line 82-83 : “out of 23 countries 18 belong to LMIC”, does this mean podocniosis is found in 5 high income countries?

Page 5, line 106 : the term ‘dirtiness’ is highly stigmatizing word and good to reconsider using such word. 

Page 7, line 172 : it is stated as ‘ there are 33 individuals belonging to traditional religion”. Do you mean only 33 individuals follow traditional religion in such parts of Africa.

Page 9, line 198 : ‘each participant gave written consent after asking questions and receiving responses’- please rephrase it, as it looks questioning or data collection was conducted before informed consent.

Page 11, line 249-256 : assigning percentage in such small counts (2-15) does not look statistically acceptable

**Summary and General Comments**

Reviewer #1: This paper addresses an important issue in NTD research. The findings presented can be translated into policy and action not only in the context of podoconiosis prevention and control, but also for other stigmatizing NTDs.

Reviewer #2: Overall, this is a timely and very well executed piece of research, adding a level of nuance to the field of health-related stigma, that is commendable and should be more widely applied to similar work in future. 

Taking into account my minor comments, I would recommend this paper for publication.

Reviewer #3: It is well designed and well conducted study with interesting outcomes

However, for publication it is too much detailed, it should be summarized.

After all this intense analysis and discussion , it is good to have clear and elaborated recommendations to alleviate podoconiosis related stigma

PLOS authors have the option to publish the peer review history of their article (what does this mean?). If published, this will include your full peer review and any attached files.

Reviewer #1: No

Reviewer #2: Yes: Julian Eaton

Reviewer #3: No

Figure Files:

Data Requirements:

Reproducibility:

References

---

## [Editor Report · Decision Letter 1]

5 Oct 2024

Dear Dr Bikorimana,

We are pleased to inform you that your manuscript 'Uncovering intersecting stigmas experienced by people affected by podoconiosis in Nyamasheke district, Rwanda' has been provisionally accepted for publication in PLOS Neglected Tropical Diseases.

Best regards,

Bruce A. Rosa

Academic Editor

Victoria Brookes

Section Editor

The authors have sufficiently addressed reviewer concerns in the revised version of the manuscript.

---

## [Editor Report · Acceptance letter]

14 Oct 2024

Dear Dr Bikorimana,

We are delighted to inform you that your manuscript, "Uncovering intersecting stigmas experienced by people affected by podoconiosis in Nyamasheke district, Rwanda," has been formally accepted for publication in PLOS Neglected Tropical Diseases.

Best regards,

Shaden Kamhawi

co-Editor-in-Chief

Paul Brindley

co-Editor-in-Chief
